# Physico-Chemical and Antimicrobial Efficacy of Encapsulated Dhavana Oil: Evaluation of Release and Stability Profile from Base Matrices

**DOI:** 10.3390/molecules27227679

**Published:** 2022-11-08

**Authors:** Shirish K. Phanse, Shriya Sawant, Harinder Singh, Sudeshna Chandra

**Affiliations:** 1Department of Chemistry, Sunandan Divatia School of Science, SVKM’s NMIMS (Deemed to be University), V. L. Mehta Road, Vile Parle (West), Mumbai 400056, India; 2Department of Biological Sciences, Sunandan Divatia School of Science, SVKM’s NMIMS (Deemed to be University), V. L. Mehta Road, Vile Parle (West), Mumbai 400056, India

**Keywords:** dhavana oil, microencapsulation, release mechanism, spray-drying, antimicrobial activity

## Abstract

Essential oils (EOs) are naturally occurring volatile aromatic compounds extracted from different parts of plants. They are made up of components like terpenes, phenols, etc., and are chemically unstable and susceptible to oxidative deterioration, leading to reduced shelf-life and overall degradation of the product. Encapsulation of EOs in a matrix can prevent degradation of the active ingredient and improve the shelf-life. In this paper, we report encapsulation of Dhavana oil (*Artemisia pellen*) in a modified starch matrix using a spray-drying technique. Physico-chemical properties of neat and encapsulated Dhavana oil were studied. We selected two powder bases: CaCO_3_ and TALC and, loaded neat and encapsulated Dhavana oil in it, studied their stability and interaction with the base matrices at 3 °C, 22 °C and 45 °C up to 2 months under closed conditions and one week at 22 °C and 45 °C under open condition. Thermal degradation pattern was studied for neat and encapsulated Dhavana oil and modified starch. Release of primary active component of neat and encapsulated Dhavana oil from the base matrices was evaluated with GCMS. Stability study and release mechanism were elucidated to understand the release pattern in different base powders under similar conditions. Retention of hydroxydhavanone was found to be better in TALC than CaCO_3_, and therefore, the former can be considered a suitable base matrix for developing a stable powder formulation with an optimum release of the oil. Dhavana oil is known for its anti-microbial activity, and hence, neat and encapsulated Dhavana oil was tested on different bacterial and fungal strains. The encapsulated oil depicted good anti-microbial efficacy against various bacterial and fungal strains, which is a step forward for developing anti-microbial formulations. Thus, the reported work will provide helpful information on cosmetic formulation and, therefore, be useful for perfumery, food, and cosmetic industries.

## 1. Introduction

Essential oils (EOs) contain many unsaturated organic compounds, such as terpenes, phenols, aldehydes, and ketones. EOs are highly volatile, due to which they cannot be stored for a long time. Storage conditions like temperature, humidity, and sunlight, may induce oxidative degradation of components that have low vapor pressure. Due to this, changes in the olfactory profile in essential oil are observed. Also, the oil may turn rancid, or its odor strength may diminish over a period. In other words, loss of volatile components from essential oils can impact the olfactory profile and strength of EOs. Heat and light can also induce chemical instability in the EOs. Mehdizadeh et al. reported better stability and retention of physical properties of cumin (*Cuminum cyminum* L.) essential oil at 4 °C and −20 °C than room temperature and above [1]. Thus, it is essential to find a technique that can preserve the olfactory properties and strength of EOs and provide chemical stability to the active components under various storage conditions.

Encapsulation is a beneficial and proven technique that can protect EOs from degradation and loss of strength and retain the olfactory properties for a longer period [2]. Encapsulation involves formation of a protective layer/shell/wall matrix around the oil so that it can prevent evaporation/faster release of volatile compounds from the matrix. The protective barrier layer also prevents exposure of EOs to harsh storage conditions, thereby preventing degradation and ultimately improving the life [3]. Once encapsulated, EOs can have improved chemical stability and retain their properties, making the formulation more sustainable.

Dhavana oil is one of the rare essential oils produced in India which is extracted from the leaves and flowers of a plant species known as “*Artemisia pallen*”. Production of Dhavana oil is about 1.5 to 2.00 metric tons. It is seasonal and, therefore, costly. Dhavana oil has a distinctive smell that can be described as fruity, berry, ripe fruity, dried fruit, apricot, wormwood, aromatic, woody, sweet and balsamic [4] and used in varying percentages in flavor and fragrance compositions. 

In the perfumery industry, it is observed that the same fragrance with the exact dosage smells differently in different base formats, i.e., the olfactive strength of fragrance oil differs in different base matrixes. This is equally true for EOs when applied to two different base materials. This is due to various components in EOs that possess different functional groups that exhibit varying vapor pressures and boiling points. Apart from chemical properties, physical interaction of oil with base matrix also varies. Composition of base material, particle size, hydrophobicity/hydrophilicity, absorption, or adsorption properties of the base matrices may influence the release pattern of essential oil. It is also reported that the olfactory characteristics and physico-chemical properties of an EO remain stable when encapsulated in a starch matrix [5]. Rakmai et al. encapsulated black pepper oil in cyclodextrin to improve stability and retention of limonene, 3-carene and pinene [6]. Encapsulated products are now used in powder formulations like detergent, talcum, and surface cleaning powder. Thermal stability of encapsulated products can be further improved by adding them into TALC or similar powder matrices and forming a premix before final use.

There are various encapsulation techniques like spray-drying, complex coacervation, and coaxial electrospinning and these processes form either micro- or nano-encapsulated products. Out of all, spray-drying is preferred over other method. Spray-drying is fast, simple, and easily controlled as per the requirement [7,8]. because the product dries faster, and the encapsulated slurry can be converted into a dried powder in a single step. In spray-drying, various compounds like biopolymers, starch, and gums are used. Renata et al. used the spray-drying method to encapsulate EO of peppermint (*Mentha piperita* L.) with n-octenyl succinic anhydride-modified starch to form free-flowing powder and lower the moisture content in the microcapsules [9,10,11]. Hu et al. used various wall material like protein isolate (WPI), maltodextrin (MD), and sodium alginate, sodium caseinate to encapsulate cinnamon EO to achieve stability of the active ingredients at 50 °C for one month [12,13]. Base matrices are selected depending on the type of essential oil, plant extracts, vitamins, polyphenols, lipase, or peptides. Selection of a proper wall matrix is essential since it impacts the encapsulated product’s chemical stability, fragrance profile, strength, release, and retention [14]. Some of the base wall matrix materials which are used for encapsulating oils are hydroxypropyl methylcellulose (HPMC), gum acacia, gum arabic, gelatin, dextrin, starch, modified starch, pectin, chitosan, and carboxymethyl cellulose (CMC) [15].

Various EOs, including Dhavana oil, exhibits antimicrobial activity. Essential oil contains lots of compounds that show antimicrobial activity, few of them are davanone (50–56%), bicyclogermacrene, dhavana ether, ethyl cinnamate, methyl cinnamate, linalool, caryophyllene oxide, phytol, spathulenol [16,17]. Thus, essential oil may become a potential source of new active molecules against resistant bacteria or fungus for the pharmaceutical industry. Similarly, it may be helpful for the food industry. Thus, it is essential to study the antimicrobial properties of EOs.

In this paper, we report encapsulation of Dhavana oil in modified starch matrix and their stability and release pattern from base matrices, i.e., TALC and CaCO_3_. Encapsulation efficiency was evaluated, and physico-chemical properties of encapsulated Dhavana oil were studied at various temperatures and time period. A modified octenyl-succinated emulsifying starch was used in the encapsulation process. Release of neat and encapsulated oil was compared under open and closed conditions at various temperatures and time period. Dhavana oil has been reported to have antiviral and antimicrobial properties and is known for its medicinal properties. Ruikar et al. checked an antimicrobial activity of an aerial part of *artemisia pallen* extracted in n-hexane, chloroform, and methanol. Only methanol extract showed activity against bacteria [18,19]. We are reporting the growth inhibition properties of Dhavana oil against gram-positive and gram-negative bacteria. Dhavana oil has effective activity against oral care bacteria [20,21]. We tested and studied the efficacy of neat and encapsulated Dhavana oil against bacterial and fungal strains. 

We used CAPSUL^®^, a modified starch (sodium octenyl succinate), to encapsulate Dhavana oil due to its excellent water-solubility properties which helps in forming emulsion quickly after dispersion. Even at a 50% concentration, it forms a flowable liquid with less viscosity. It has both hydrophilic and hydrophobic properties; and therefore, it exhibits strong oil retention properties. Besides high oil retention ability, it also has good wall forming properties, that contributes to process efficiency and product quality by extending shelf-life of a material. Further, modified starch is readily available, biodegradable, easy to handle and, above all, they originate from natural sources. 

TALC is a common filler material used in cosmetic preparation, such as talcum powder, soap, face packs, mascara, baby powder, lipstick, etc. It originates from natural sources and is safe on the skin. TALC is a good absorbent for oils, fragrances, and essential oils, and is envisaged to play an important role in fragrances and encapsulated product retention. Understanding the interaction of TALC with fragrance or essential oil is necessary to formulate powder formulations, such as talcum powder, cleaners, detergents, etc. On the other hand, CaCO_3_ is used as a filler material in many cosmetic formulations like toothpaste, powder, and lotions. It is added as a whitener or an opacifier and also has good oil absorption properties. However, it is important to study the stability of essential oils in the filler materials. TALC and CaCO_3_ exhibit different chemical and physical nature, shapes, sizes, and morphology. Therefore, it is essential to understand the interaction of neat and encapsulated Dhavana oils with TALC and CaCO_3_ and find out the more suitable filler material for the fragrance.

## 2. Materials and Methods

### 2.1. Materials

Dhavana oil was procured from Leela Aromatics Ltd. Mumbai, India (GCMS analysis shows the presence of sixty-one components, majorly 56.32% davanone, 6.87% bicyclogermacrene, and 5.6% ethyl cinnamate). CapsuleTM Gum, i.e., sodium octenyl succinate, i.e., modified starch, was procured from Ingredions Inc., Westchester, IL, USA. Tri potassium citrate and sucrose were procured from Merck Ltd, Mumbai, India, Citric acid anhydrous 99.00%LR grade and calcium carbonate special grade 98.5–100% were purchased from SDFCL. 85% AR Hexane, 99.80% methanol, and TALC extra pure were purchased from Loba Chemie Pvt. Ltd. Mumbai, India. Sipernat Silica Powder D-17 was obtained from Evonik India Pvt. Ltd., Mumbai, India.

### 2.2. Analytical Methodology

Following methods and equipment were used for preparing the emulsion and carrying out the physico-chemical analysis of neat and encapsulated Dhavana oil. Physical parameters like color and odor were checked visually and smelling on a smelling strip, respectively. Solubility of the oil was checked in hexane and ethyl alcohol. Abbe refractometer (Atago) was used to measure the refractive index. Titrimetric method was used to measure the acid value/saponification value. Flash point and boiling point were measured by Seta Multiflash (closed cup) and capillary method, respectively. 

The Silverson high shear mixer blade (Tubular assembly with a speed of 10,000 rpm) was used to blend, emulsify, homogenize, hydrate, and dissolve Dhavana oil and modified starch emulsion before spray-drying (Silverson Machines, Inc., East Longmeadow, MA, USA). Spray dryer from JISL (Jay Instruments & Systems Pvt Ltd., Navi Mumbai, India) of evaporation capacity 1L/hr of water was used for spray-drying. The drying air temperature was set at 160–170 °C with a heater capacity of 2.5 KW. Compressed air for spray flow was maintained at 2.5 bar. The spray dryer chamber diameter was 4″, 6″, fitted with two-fluid nozzle apertures. Microstructural analysis of the encapsulated product was analyzed using a scanning electron microscope (JEOL JSM-7600 FEG-SEM) at 15 kV. The accelerating voltage was in the range of 0.1 to 30 kV. The particle size analyzer Malvern Mastersizer (Malvern Panalytical Ltd., Malvern, UK) was used to measure the particle size of the encapsulated product. For quantitative and qualitative analysis of active ingredients, gas-chromatograph coupled with a mass spectrometer (GC-HRMS) with the following specifications was used: Make of MS: Jeol, Model: Accu. TOF GCV-Specification: EI/CI Source: Time of Flight Analyzer/Mass range −10–2000 amu/Mass resolution −6000. Make of GC: Agilent 7890, FID detector: Head Space injector: Combipal autosampler. Agilent -model no 6890-G1530N GC and 5973-G2579A-MS. Gas type used as a carrier- Helium, Detector type-MS, volume approximate used for injection-1 microliter (Agilent Ltd., Santa Clara, CA, USA).

pH meter (Oakton- Cole Parmer India) was used to measure pH of the solutions, and Brookfield viscometer Model-RVDV-I+ was used to measure viscosity of the emulsion. Thermal gravimetric analysis was performed on Perkin Elmer thermal analyzer. Measurement was done with a heating rate 10 °C per min from 25 °C to 475 °C. For stability studies, ovens of Thermolab Ltd., Vasai-Virar, India (with 1 °C +/−variation) were used. Studies were conducted at various conditions like 3 °C, 22 °C (RT), 45 °C, and at 37 °C with 70% RH. 

The Clevenger hydro-distillation method extracts essential oils and bioactive compounds from encapsulated products to estimate the total oil. Moisture content was determined by the Karl Fisher method. 

FTIR-(Shimadzu)-IRAFFINITY-1S was used to record FTIR spectra of neat and encapsulated Dhavana oil, CaCO_3_, TALC, and modified starch. The experimental details are as below.

### 2.3. Experimental Details

#### 2.3.1. Physio-Chemical Analysis of Dhavana Oil

The neat Dhavana oil was analyzed by GCMS to check the main active (in this case, davanone) and other constituents. Appendix A shows the GC-MS analysis of the neat Dhavana oil. 

#### 2.3.2. Formation of an Emulsion of Dhavana Oil

To form an emulsion of Dhavana oil, citric acid, sucrose, and tri-potassium citrate were mixed in DM water in a 200 mL beaker and stirred at ~3500 rpm to dissolve all the components thoroughly [22]. Modified starch was used as wall material with sucrose. Modified starch was taken in a dried petri dish and added slowly to the above solution to form dispersion without any lumps. Complete dispersion was achieved in 25–30 min. A Silver-son mixer was used to mix the dispersion, keeping the stirring speed at 5500–6000 rpm. After dispersion, high-speed stirring (6000 rpm) was continued for 5–6 min. After that, stirring was stopped, and the container was sealed with stretched PP foil and Aluminum foil. The dispersion was allowed to stand overnight, and consistency of the dispersion was checked after 24 h. The dispersion was found to be stable and transparent, with no layer separation.

Following day, neat Dhavana oil was taken in a separate 50 mL glass beaker. Dispersion of modified starch was again kept under stirring at a speed of 3000 rpm. Dhavana oil was added to the dispersion and homogenized at 6500–7000 rpm for 5–6 min at 25 °C. A pure milky white emulsion was obtained without any layer separation. The process was repeated thrice to ascertain the stability of the emulsion. The emulsion was tightly sealed with stretched PP foil and aluminum foil to avoid evaporation losses and kept under observation overnight. The emulsion was checked for its consistency after seven days and was found to be stable (Appendix A). Viscosity was measured at 25 °C using Brook field viscometer and was found to be 60 cps. A pH of 4.57 was recorded for the emulsion. Several trials were performed to optimize encapsulation analyzed by SEM and moisture content. The encapsulation process produced good, free-flowing, oval to round shape capsules. Table 1 lists the amounts of various ingredients used to prepare the emulsion. Using below formulation, we proceed for an encapsulation process.

#### 2.3.3. Encapsulation of Dhavana Oil

Spray-drying of the emulsion was carried out to encapsulate the Dhavana oil by reported methods. Gomez et al. used various combination of native rice starch, modified rice starch, maltodextrin, hydrolyzed protein to encapsulate orange essential oil via spray-drying method. Modified starch when used, more than 50% produced stable, free-flowing microcapsules with increased encapsulation efficiency [23,24]. Table 2 shows the parameters set in the spray dryer. The stable emulsion was fed through a feeder into the atomizer, which dispersed the liquid or slurry in controlled drop size. Hot drying air was passed as a co-current source. Specific nozzle size was selected to get particles in the range of 5 to 200 µm, and the product was obtained in a cyclone. The dry powder of encapsulated Dhavana oil was free-flowing and slightly yellowish. The yield of the product was ~90%. 

Various factors like percentage and type core oil percentage, polymer or wall material, inlet-outlet temperature, emulsifiers and emulsion stability, core-to-wall material ratio etc. govern the morphology of the final encapsulated product. The inlet and outlet temperature are important to remove a moisture from a slurry in co-current spray-drying process. Temperature influences the moisture content, density, and free flowing nature of a microcapsule. In co-current process, an emulsion of Dhavana oil in water as a flowable slurry was subjected to spray dryer. When emulsion droplet come in contact with hot air, due to hot inlet flow, droplet start losing moisture very fast and at the same time encapsulation is completed due to presence of wall material i.e., modified starch. Core oil becomes encapsulated within a modified starch and forms a microcapsule, which is finally collected as a product. Since the encapsulation process becomes completed and Dhavana oil is well protected within the wall material, no further losses of oil are observed. Thus, a good encapsulation efficiency can be achieved at inlet/outlet temperature set at 172 °C and 92 °C respectively. This range of temperature restricts the moisture content below 2%.

#### 2.3.4. Evaluation of Total Oil Content 

For evaluation of total oil content [11] in the encapsulated product, 2.0228 g of encapsulated Dhavana oil was added to 150 mL DM water in a 500 mL round bottom flask. It was mixed well to dissolve all the encapsulated product. A round bottom flask was attached to Clevenger’s apparatus. Additional 50 mL DM water was added from the top of the condenser. The mixture was refluxed for 2 hrs. After a while, the essential oil got collected inside the condenser arm. Volume of the collected oil was measured and calculated as per the below equation. The Clevenger’s apparatus was calibrated with water levels in ml before the experiment.

P_sd_ = Percentage (*v*/*w*) of Steam distilled oil.P = Real oil content in an Encapsulated product (%*v*/*w*)V_sample_ = Final volume in ml of oil collected in a sidearm of Clevenger’s apparatus.X = Weight of encapsulate sample taken before steam distillation. i.e., sample weight.

Density × Volume = Mass,
$$P_{\mathrm{sd}}=\frac{V_{\mathrm{sample}}}{X}\times 100\;\ldots \left(\%v/w,\;V=\mathrm{volume}\times \mathrm{density}\right)$$

#### 2.3.5. Evaluation of Surface Oil

0.501 g of encapsulated Dhavana oil sample was taken in a small round bottom flask, and 10 g hexane was added to it. The flask was shaken, and the solution was filtered through a Whatman filter paper no. 41, and the residue was collected. The filtrate was collected in a previously weighed glass petri dish (W_1_) and was kept in a water bath to evaporate hexane. The glass petri dish was again weighed (W_2_), and the difference in weight of the Petri dish was calculated.
$$\mathrm{Surface}\;\mathrm{Oil}=\frac{W_{2}-W_{1}}{\mathrm{Actual}\;\mathrm{Total}\;\mathrm{oil}\;\mathrm{in}\;\mathrm{encapsulate}}\times 100$$
Encapsulation and entrapment efficiency was calculated using the formula given below:$$\;\mathrm{Encapsulation}\;\mathrm{Efficiency}\;\left(\%\right)=\frac{\mathrm{Total}\;\mathrm{oil}\;\mathrm{experimental}\;\mathrm{loading}\;\left(g/g\right)\;\mathrm{powder}}{\mathrm{Theoretical}\;\mathrm{loading}\;\left(g/g\right)}\times 100$$
$$\;\mathrm{Entrapment}\;\mathrm{Efficiency}\left(\%\right)=\frac{\left(\mathrm{Total}\;\mathrm{oil}\;\left(g/g\right)\;\mathrm{powder}\right)-\left(\mathrm{Surface}\;\mathrm{oil}\;\left(g/g\right)\;\mathrm{powder}\right)}{\mathrm{Total}\;\mathrm{oil}\;\left(g/g\right)\;\mathrm{powder}}\times 100$$

#### 2.3.6. Thermal Analysis

Thermal degradation pattern and release by % weight loss of encapsulated Dhavana oil were evaluated using TGA. Measurements were carried out between 25 °C to 450 °C, and the heating rate was 10 °C per minute. It is assumed that neat Dhavana oil will degrade or evaporate more rapidly than encapsulated Dhavana oil. In case of encapsulation, modified starch played a significant role in retaining Dhavana oil into its matrix. The degradation profile of modified starch gum was also studied. A graph of weight loss against temperature change was plotted. 

Thermal degradation and % weight loss of neat and encapsulated Dhavana oil were evaluated in two powder bases, i.e., TALC and CaCO_3_. Since both the bases have different chemical and physical properties, particle size, absorption, and adsorption capacity, it is hypothesized that the degradation and release profile of neat and encapsulated Dhavana oil will be different in both the base materials. Following samples were prepared and subjected to TGA, and the results are shown in Appendix A.

Batch no. 001: Neat Dhavana oil (1% loading) in TALC.Batch no. 002: Encapsulated Dhavana oil in TALC (3.35% loading as encapsulated Dhavana oil contains a 30% neat oil)Batch no. 003: Neat Dhavana oil (1%) in CaCO_3_.Batch no. 004: Encapsulated Dhavana oil in CaCO_3_ (3.35% loading as encapsulated Dhavana oil contains a 30% neat oil)

Release of neat and encapsulated Dhavana oil from powder base TALC and CaCO_3_ was monitored under various conditions by GCMS (using extraction of Dhavana oil in a hexane solvent). It is expected that specific experimental conditions like elevated temperature may lead to changes in strength and olfactive profile of EOs due to heat, oxidation of active components, loss of low vapor pressure compounds, etc. Turek et al. emphasized on temperature being one of major factors to impact chemical properties of EOs. At higher temperature, oxidation may affect the chemical properties of the components of EOs [25,26]. Therefore, loss of strength and release profile of neat and encapsulated Dhavana oil were studied. We expected that the loss of neat Dhavana oil might be higher than the encapsulated Dhavana oil. After every month, samples stored at 45 °C were removed and subjected to analysis. A fixed quantity of 2.384 g sample was added to 10 mL of DM water and mixed well with a stirrer for 10 min. In this process, modified starch gets dissolved in water, and Dhavana oil gets released in water, which was extracted using hexane, and after extraction, the solution was concentrated to 8 g. The hexane extract was then injected into GCMS. Following samples were prepared:
Neat Dhavana oil @ 1% in TALC and CaCO_3_.3.35% of encapsulated Dhavana oil in TALC and CaCO_3_. Encapsulated oil contains 30% oil loading, i.e., (3.35 × 30% = 1.005), to keep the oil ratio approximately the same in both formats. For stability studies in TALC 
Batch no. 001/1–Neat Dhavana oil in TALC for 45 °C/1 monthBatch no. 001/2–Neat Dhavana oil in TALC for 45 °C/2 monthBatch no. 002/1–Encapsulated Dhavana oil in TALC for 45 °C/1 monthBatch no. 002/2–Encapsulated Dhavana oil in TALC for 45 °C/2 monthFor stability studies for the CaCO_3_
Batch no. 003/1–Neat Dhavana oil in CaCO_3_ for 45 °C/1 monthBatch no. 003/2–Neat Dhavana oil in CaCO_3_ for 45 °C/2 monthBatch no. 004/1–Encapsulated Dhavana oil in CaCO_3_ for 45 °C/1 monthBatch no. 004/2–Encapsulated Dhavana oil in CaCO_3_ for 45 °C/2 month

It is hypothesized that at higher temperature, encapsulated Dhavana oil will be able to retain the active ingredients, unlike neat oil, because of the protection from the modified starch [24,25]. To prove this hypothesis, the following experiment was conducted. 

The sample was divided into two groups. In one group, sample was taken in a Petri dish and kept at RT (in fume hood), while in the other group, the sample was kept in an open Petri dish at 45 °C (inside stability oven). Both the samples were kept for seven days, and then the samples were evaluated for an olfactory profile by a panel of 5 team members. Initial assessments revealed that the panelists could not smell any trace of Dhavana oil. Then the sample was also subjected to GCMS analysis, wherein 1.950 g sample was added to 10 mL of DM water and stirred well for 10 min. The essential oil content was extracted from hexane and subjected to GCMS analysis. The following sets were taken for analysis:

Batch no. 001A–Neat Dhavana oil in TALC for 22 °C (RT), one-week, open petri dish;Batch no. 001B–Neat Dhavana oil in TALC for 45 °C, one-week, open petri dish;Batch no. 002 A–Encapsulated Dhavana oil in TALC for 22 °C (RT), one-week, open petri dish;Batch no. 002B–Encapsulated Dhavana oil in TALC for 45 °C, one-week, open petri dish;Batch no. 003A–Neat Dhavana oil in CaCO_3_ for 22 °C, (RT) one-week, open petri dish;Batch no. 003B–Neat Dhavana oil in CaCO_3_ for 45 °C, one-week, open petri dish;Batch no. 004A–Encapsulated Dhavana oil in CaCO_3_ for 22 °C (RT), one-week, open petri dish;Batch no. 004B–Encapsulated Dhavana oil in CaCO_3_ for 45 °C, one-week, open petri dish.

#### 2.3.7. Sensory/Olfactive Analysis 

Pereira et al. reported olfactive analysis or sensory values for odor, which are relative measures of specific fragrance and intensity. The odor intensity of any fragranced product is affected by the base matrices. Every base has its own odor and intensity as well as every base made up of different constituents in it. It shows different physical attraction forces due to absorption and adsorption [27]. Therefore, sensory/olfactive analysis was performed to analyze the odor of neat and encapsulated Dhavana oil. Effect of temperature on release of encapsulated Dhavana oil from TALC and CaCO_3_ was evaluated by olfactive analysis. The experiment evaluated the interaction of neat and encapsulated Dhavana oil with a base material. Neat and encapsulated Dhavana oil was loaded in TALC and CaCO_3_, and their interaction with the base was evaluated after one-, two- and three- months at various temperatures. 

The evaluations were carried out by the following measurements. 

The intensity scale of neat v/s encapsulated Dhavana oil in both the bases;Intensity scale of neat Dhavana oil in TALC v/s neat Dhavana oil in CaCO_3;_Intensity scale of neat Dhavana oil in TALC and CaCO_3_ v/s encapsulated Dhavana oil in TALC and CaCO_3_ when dissolved in water solution.

Neat and encapsulated Dhavana oil in TALC and CaCO_3_ were subjected to stability checks at varying experimental conditions: 3 °C, 22 °C (RT), 37 °C and 70% RH, 45 °C for 1/2/3 months (Appendix A). Samples were packed in 30 g glass bottles with a white HDPE cap with an inner wad. Three sets of each sample were kept for every condition. Olfactive evaluation of the samples was done with an intensity scale of 1 to 10 by eight olfactive evaluation experts from Firmenich Lab. The intensity scale measures the odor strength of the product (Appendix A). Carranza et al. used a sensory or olfactive analysis method to evaluate the fragrance of orange essential oil. Eight different formulations were tested by ten different fragrance experts to find out best performing formulation [28]. Evaluation experts were asked to rate the odor strength of neat and encapsulated Dhavana oil every month. Readings at 3 °C, 22 °C, (RT), 37 °C and 70% RH, 45 °C were recorded. Main focus was on the results of 45 °C since the temperature was obtained by keeping the samples in a dried oven, and it was free from any moisture. In 3 °C, 22 °C, (RT), 37 °C, and 70% RH, humidity level was high compared to 45 °C. Higher humidity can disintegrate the encapsulation, leading to varying release profiles. A web diagram was plotted after three months, representing the oil’s olfactive profile in different powder bases. Subsequent batches were made:Batch no.-001-Dhavana oil in TALC-oil loading @ 1.00%;Batch no.-002-Encapsulated Dhavana oil in TALC-loaded at 3.35% (encapsulation at 30% of oil);Batch no.-003-Dhavana oil in CaCO_3_ -oil loading @1.00%;Batch no.-004-Encapsulated Dhavana oil in CaCO_3_ -loaded at 3.35% (encapsulation at 30% of oil).

We tested the sample of TALC and CaCO_3_ with neat and encapsulated Dhavana oil in water to check the intensity impact at every month interval. 0.20 g sample from each of the above batches was taken in a 100 mL beaker, and 50 mL water was added to it. Samples were stirred to dissolve the product completely and sent to all the panelists, and intensity was recorded. 

#### 2.3.8. Fourier Transform InfraRed (FTIR) Analysis 

FTIR analysis of CaCO_3_, TALC, neat and encapsulated Dhavana oil, modified starch, and neat and encapsulated Dhavana oil loaded in CaCO_3,_ and TALC was done to elucidate the interaction of the oil with CaCO_3_ and TALC through changes in the functional group’s signals. 

#### 2.3.9. Antimicrobial Testing

##### Preparation of Dhavana Oil Stock Solution for Antibacterial and Antifungal Study

Sabouraud Dextrose agar and Meuller Hinton agar were used for antifungal and antibacterial tests, respectively. Agar well diffusion method was used to prepare the stock solution of the oil, which is mentioned below briefly.

Neat Dhavana oil stock solution was prepared by mixing 1 part of Dhavana oil with 2 parts of solubilizer (Cremophor RH40). Therefore, 500 mg (540 µL) Dhavana oil was mixed with 1.0 g (980 µL) Cremophor RH 40 to make a stock solution of 328.0 mg/mL. concentrations of Dhavana oil. Four concentrations were tested against standard A = 300.0 mg/mL, B = 150.0 mg/mL, C = 75.0 mg/mL, D = 37.5 mg/mL.

Using the prepared stock solution, dilutions were prepared using St. saline as a diluent to give the following concentrations of Dhavana oil.

A = 300.0 mg/mL, B = 150.0 mg/mL, C = 75.0 mg/mL, D = 37.5 mg/mL.

##### Preparation of Encapsulated Product Stock Solution

Encapsulated product stock solution was prepared to obtain 300 mg/mL oil to compare results with neat Dhavana oil. For the same, 1.12 g encapsulated product was dissolved in 1 mL St. warm saline for complete solubilization. The stock consisted of 300 mg/mL Dhavana oil. Using the prepared stock solution, dilutions were prepared using st. saline.

##### Growth and Maintenance of Bacterial Strains

*Escherichia coli*, *Staphylococcus aureus*, *Micrococcus* spp., *Bacillus subtilis*, *Serratia* spp., *Klebsiella pneumoniae*, *Streptococcus mutans* ATCC 25175, *Salmonella typhi* and *Pseudomonas aeruginosa* were revived from their respective glycerol stocks. These bacterial strains are part of the culture collection of the Department of Biological Sciences, Sunandan Divatia School of Science, NMIMS University, Mumbai. S. mutans was grown using st. Brain Heart Infusion broth for 48 h under anaerobic conditions at 37 °C. All other strains were grown in st. Tryptone Soya Broth for 24 h at 37 °C.

##### Antibacterial Agar Well Assay

All antimicrobial assays were performed using st. Meuller Hinton agar. All the bacterial cultures (except *S. mutans*) were adjusted to 0.1OD, whereas *S. mutans* were adjusted to 0.2 OD @600 nm using BioTek EPOCH2 microplate reader. Pour plate was carried out using 200 µL of OD adjusted cultures, which were added to 20 mL molten and cooled st. Meuller Hinton agar butts. The butts were mixed well and poured to st. petriplates and allowed to cool. 8 mm wells were bored in the agar using st. cork borer and 50 µL prepared dilutions were added to the wells, followed by incubation at 37 °C for 24 h (under anaerobic conditions for *S. mutans*). After 24 h incubation, inhibition zone sizes were measured and tabulated. Assays were performed in triplicates, and the zone of inhibition was recorded as mean ± standard deviation (SD) values of three replicates.

##### Minimum Inhibitory Concentration and Minimum Bactericidal Concentration Assay

The antimicrobial activity of the encapsulated Dhavana oil was evaluated by calculating their minimum inhibitory concentration (MIC) and minimum bactericidal concentration (MBC) against Gram-positive Staphylococcus aureus and Gram-negative Escherichia coli. The MIC value for both the strains was determined by the broth microdilution method. A series of two-fold dilution of encapsulated Dhavana oil ranging from 32–0.25 mg/mL was dispersed in the columns of wells. The stock of Dhavana oil was prepared using St. saline, and the dilutions were prepared using double strength Mueller-Hinton broth. Finally, 20 µL of bacterial suspension corresponding to 10^4^ cells/mL was inoculated in each well and incubated at 37 °C for 24 h. Turbidity measurement was performed after incubation visually and in a microplate reader (Epoch2 BioTek Microplate reader) at 595 nm. The MIC (mg/mL) was defined as the lowest concentration tested, which did not allow cell growth within 24 h at 37 °C.

For observing MBC, 10 µL of the medium from wells was transferred to a Tryptone soyabean agar plate and incubated at 37 °C for overnight. The MBC was determined as the lowest concentration (μg/mL) where no growth was observed. All determinations were performed in triplicate to ensure the reproducibility of the results.

##### Antifungal Testing

Antifungal activity was checked against *Trichoderma virens* NCIM1298, *Penicillium funniculosum* MTCC 2552, *Chaetomium globosum* MTCC 2193, Aspergillus niger NCIM 596, *Aspergillus brasiliensis* NCIM1025, *Penicillium citrinum* MTCC 2547 and *Aureobasidium pullulans* NCIM 1049 was carried out using St Sabouraud Dextrose agar. Undiluted Sterile Agar was cooled and seeded with the test organism. Plates were poured and allowed to solidify. Wells (8 mm) were bored with the help of cork borer. 200 μL of test sample concentrations was put into the wells. Plates were incubated at 28 ± 1 °C for 5–7 days. After incubation, plates were observed for a zone of clearance/inhibition, and an assay was performed in triplicate.

## 3. Results and Discussion

### 3.1. Physico-Chemical Properties of Neat Dhavana Oil

Detailed physico-chemical analysis of the neat Dhavana oil is presented in Table 3. 

### 3.2. Physico-Chemical Properties of Encapsulated Dhavana Oil

To obtain a free-flowing encapsulated product, inlet temperature of the spray dryer was increased to 170 °C, and 125 g of the emulsion was subjected to spray-drying. Approximately 40 g of encapsulated product was obtained. Encapsulation of Dhavana oil was calculated to be 30%. Moisture content of encapsulated Dhavana oil was found to be 0.631%. Scanning electron microscopic images showed round/oval-shaped capsules ranging from 5 to 200 microns (Figure 1). 

Particle size of the encapsulated oil was also studied [11,13,15] and presented in Table 4. Average particle size (psd) was found to be ~60 µm, and d (0.90) value < 59.36 µm as calculated from the dynamic light scattering technique (DLS). The particle size distribution is shown in Appendix A. 

Kausadikar [8] encapsulated lemon oil separately in maltodextrin, gum Arabic and modified starch. The particle sizes of the microcapsules were in the range of 10 to 14 μm. Increase in the amount of modified starch showed improved encapsulation and entrapment efficiency of the oil. On the other hand, Hermanto et al. [15] found that when cinnamon oil was encapsulated with a different combination of maltodextrin and a gum arabic in a ratio (0:1, 1:0, 1:1, 2:3), the moisture content increased from 5.60% to 7.71% and the content of surface oil decreased from 4.20% to 2.60%. The particle size ranged from 1.92 µm to 30.80 µm.

The content of total oil and surface oil in an encapsulated product provides information on the extent of oil entrapped. Table 5 shows the total oil/surface oil content and encapsulation efficiency of Capsule^TM^ Modified Starch. Experiments were done in triplicate. 30% Dhavana oil was used for encapsulation. After encapsulation, the total oil content was calculated to be 29.02% which depicts the actual retention of Dhavana oil in the microcapsules after encapsulation. On the other hand, the surface oil was found to be 3.526%. The encapsulation efficiency of essential oil is dependent upon the surface oil and are used as a quality parameter. Lower the surface oil, better is the quality with respect to encapsulation and entrapment efficiency of the product. The microcapsules of Dhavana oil showed the encapsulation efficiency of 96.46% and entrapment efficiency of around 87.84%.

We conducted a Thermogravimatric analysis to evaluate the thermal stability of Dhavana oil, Modified starch, and encapsulated Dhavana oil. Similarly, we studied the stability of neat and encapsulated DO in TALC and CaCO_3_ powder base.

#### 3.2.1. Thermal Analysis

Thermogravimetric analysis (TGA) was performed to evaluate the % weight loss of Dhavana oil, starch and encapsulated Dhavana oil and understand the degradation pattern. [29] The thermal degradation pattern and %weight loss of neat and encapsulated Dhavana oil in starch gum are represented in Figure 2.

TGA of neat Dhavana oil showed significant weight loss of ~99% from 142 to 417 °C. At 142 °C, the weight loss was ~6.8% which increased to 82% at 217 °C. Around 10% weight loss was seen from 217 to 412 °C, and the encapsulated Dhavana oil showed ~77% weight loss at 415 °C. Multi-step weight loss was observed in the TGA graph of the encapsulated oil, *viz*., around 6%, 70%, and 77% weight loss at 145 °C, 320 °C and 415 °C, respectively. Thermal analysis of encapsulation material, i.e., the modified starch, showed early weight loss at 60 °C, followed by 10% weight loss at 145 °C and 85% at 415 °C. Data reveals that the modified starch matrix provided thermal stability to the encapsulated Dhavana oil, unlike the neat oil, which degraded at ~150 °C.

Dhavana oil contains almost 55–60 ingredients, few of them are very volatile. The boiling point of Dhavana oil was found out to be 190 °C. At 142 °C, highly volatile ingredients started losing from oil as evident from the initial % weight loss. The emulsion of Dhavana oil in water was formed as a flowable slurry, which was subjected to spray dryer at an inlet temperature of 170 °C. During this process, Dhavana oil was encapsulated within the starch wall material, and therefore, degradation was prevented. An inlet temperature of 170 °C was essential to keep the moisture below 2.00% which is an important step for providing a stable encapsulated product.

Thermal degradation profile of neat and encapsulated Dhavana oil in TALC and CaCO_3_ was also evaluated, and the results are presented in Figure 3. TGA analysis of 1% loading of neat oil in TALC showed a sharp weight loss from 120–155 °C, and weight loss of 3.35% of encapsulated Dhavana oil in TALC was only 3% in the range of 170–417 °C. The first weight loss of 1.40% was observed around 170 °C, followed by another 1.50% from 288–417 °C. In case of CaCO_3_ base matrix, neat oil degraded at 97 °C, while the encapsulated oil showed multi-step degradation with a total weight loss of 3% from 113–265 °C. Above results showed that neat and encapsulated Dhavana oil remained protected in the TALC base better than CaCO_3_. It may be presumed that the layered structure of TALC provided a shielding effect from heat compared to CaCO_3_. Due to this, neat and encapsulated Dhavana oil showed better stability in TALC than CaCO_3_. This can be explained by the fact that the plate-like structure of TALC provides more surface area than the cubical CaCO_3_. TALC exhibits higher hydrophobicity than CaCO_3_ and therefore entrapped oil encapsulated inside the inner spaces in their plates. The outer layers of TALC molecules provide a heat-shielding effect to the encapsulates, unlike CaCO_3_ where the oil encapsulates gets absorbed only on the surface and may get exposed to heat more readily.

The stability of the neat and encapsulated DO was conducted at 45 °C to find out the loss of active content after 2 months by GCMS, results of which are presented below.

#### 3.2.2. GC-MS Analysis

GCMS is widely used to analyze the composition of essential oil and study the efficacy of the oil. Various components of the essential oil of *Aethionema sancakense* were characterized by GC/GC-MS wherein Linoleic acid, α-humulene, camphene and heptanal were found to be present in major amount [30]. GCMS profile of neat and encapsulated Dhavana oil in TALC after one-month storage at 45 °C showed presence of hydroxydavanone as active content @ RT-44.0583 @Area pact 30.1827% and @ RT-44.0395@Area pact 30.8049, respectively (Appendix A). After two months of storage, neat oil showed the RT of Hydroxydavanone at 44.0453@Area pact- 23.5616, while the encapsulated oil showed the presence of hydroxydhavanone at @ RT-44.0364@Area pact-26.7507 (Appendix A). It can be seen that Dhavana oil in encapsulation format showed better retention than that of neat oil.

Table 6 shows the consolidated GCMS profile of the neat and encapsulated Dhavana oil. It can be observed that Dhavana oil remained well protected in encapsulated form in CaCO_3_ as well for two months at 45 °C (Appendix A).

The release profile shows that loss of neat and encapsulated Dhavana oil in CaCO_3_ was more than TALC after one month (Table 7). The %hydroxydhavanone was marginally higher in TALC than CaCO_3_ after 2 months.

The above results show that the oil holding capacity of TALC was higher than CaCO_3_, probably due to particle size, structure, and nature of TALC [31]. Arsoy et al. studied the oil absorption property of matrix material, which depends on the surface area, and pore volume of the adsorbent. More the surface area and pore volume, higher will be the absorption efficiency [32]. The oil escaped faster from CaCO_3,_ which may be attributed to better oil absorption or adsorption on the surface of TALC than CaCO_3._ As mentioned under Section 3.2.1, the horizontal plate-like and trioctahedral layered morphology of TALC provides more surface area than the orthorhombic cubical CaCO_3_. The TALC has more intramolecular spaces between the plates than CaCO_3,_ where the cubical form has three unequal axes at right angle to each other, thereby reducing the intramolecular spaces. Due to the platy surface of the TALC, entrapment of the oils was easier and more efficient as compared to CaCO_3_. Thus, TALC prevented the heat degradation of the oil and provided more stability.

Release study of neat and encapsulated Dhavana oil from TALC (Appendix A) and CaCO_3_ (Appendix A) was carried out by exposing the samples for 1 week in open condition (Petri dish) at RT (22 °C) and 45 °C. GCMS was carried out to check the presence of hydroxydhavanone and Table 8 shows a consolidated result.

Olfactive analysis and stability studies were conducted to understand retention of neat and encapsulated Dhavana oil in TALC and CaCO_3_ at 45 °C for 3 months. The olfactive strength of the oils in both powder formats was evaluated by experts in a rating scale of 1–10, results of which are given below.

#### 3.2.3. Stability Study and Olfactory Analysis

Stability studies and olfactory analysis of neat and encapsulated Dhavana oil in TALC and CaCO_3_ were carried out as per the experimental conditions mentioned in Appendix A, and the results are given in Figure 4a–f.

Olfactive analysis shows that at 45 °C, intensity score for encapsulated Dhavana oil in TALC and CaCO_3_ was less than the neat oil in both the base matrix. Thus, it supports that the encapsulated Dhavana oil is more stable than neat oil after one-, two-, and three months. Olfactive analysis and radar plot show that at 45 °C, intensity score of encapsulated Dhavana oil in TALC was less than the neat oil in the first, second, and third months. Intensity score of encapsulated Dhavana oil in CaCO_3_ was less than the neat oil for the first month, but the same was not similar but higher in the second and third months. Comparison of intensity scores of neat Dhavana oil in TALC and CaCO_3_ showed in Table 9 below. A higher intensity score of encapsulated Dhavana oil in CaCO_3_ showed leakage of capsules. It supports that encapsulated Dhavana oil is more stable than neat oil in TALC compared to CaCO_3_ after three months.

Intensity score of neat Dhavana oil in TALC is higher than CaCO_3_ after 3 months, suggesting that the retention of neat Dhavana oil is better in TALC than CaCO_3_, indicating better TALC stability. On the other hand, intensity score of encapsulated Dhavana oil in TALC and CaCO_3_ when dissolved in water was higher than neat Dhavana oil loaded in TALC and CaCO_3_. Results proved that under stability conditions at 45 °C, encapsulated Dhavana oil remained protected inside a modified starch matrix for three months.

#### 3.2.4. FTIR Analysis

FTIR spectra of neat, encapsulated Dhavana oil and modified starch are represented in Figure 5a. It can be seen that the neat oil exhibited multiple peaks in the region of 1700–1750 cm^−1^, which is due to the C=O stretch of the ketone in davanone. The C-H stretch at 2900–3000 cm^−1^ is attributed to the backbone of the carbon-carbon chain. Modified starch showed a band around 3200–3300 cm^−1,^ which is attributed to the O-H stretching. The asymmetric stretching vibration of a carboxyl group appeared around 1500–1550 cm^−1^. After encapsulation of the oil, intensities of the peaks reduced significantly. The characteristic absorption peak of modified starch was observed at the same stretch in encapsulated product suggesting that there is no change in the modified starch after encapsulation [33]. FTIR of CaCO_3_ showed peaks at 1797 cm^−1^ (C=O), 2360, and 2972 cm^−1^ (due to C-H stretching). TALC showed sharp peaks at 450–460 cm^−1^ (MgO), 650–675 cm^−1^ (MgO/MgOH), ~1000–1050 cm^−1^ (SiO) and 3676 cm^−1^ (-C-H) (Figure 5b).

#### 3.2.5. Antimicrobial Activity of Dhavana Oil and Its Encapsulated Product

Testing of Dhavana oil on different bacterial and fungal strains was completed within 24 or 48 h (Table 10). Actively growing cultures were used to perform the assay for accurate results. A preliminary assay to determine the antibacterial property of Dhavana oil was performed on all the bacterial cultures (*Escherichia coli*, *Staphylococcus aureus*, *Micrococcus* spp., *Bacillus subtilis*, *Serratia* spp., *Klebsiella pneumoniae*, *Streptococcus mutans*, *Salmonella typhi*, and *Pseudomonas aeruginosa*). Four bacterial strains (*S. aureus*, *E. coli*, *Micrococcus* and *S.mutans*) showed better antibacterial activity against neat oil and therefore were used to test the antibacterial activity of the encapsulated product. There was no significant difference in the antibacterial activity between neat oil and encapsulated product observed against *S. aureus*, *E. coli*, and *Micrococcus* sp. Evidently, increased antibacterial activity was observed in encapsulated product against *S. mutans*.

Following agar-well method for testing antibacterial activity, MIC and MBC assays were performed. The MIC values observed for *E. coli* and *S. aureus* against encapsulated product were 16 mg/mL and 8 mg/mL, respectively. Following MIC, the MBC for *E. coli* and *S. aureus* were found to be 32 mg/mL and 16 mg/mL, respectively. This confirms the antibacterial activity of Dhavana oil is well retained after the processes required to formulate the encapsulated product. Moreover, since material used for encapsulation, i.e., CAPSUL^®^, has not been reported to possess any antimicrobial activity, hence no enhancement or decreased antimicrobial activity was observed in encapsulated product as compared to essential Dhavana oil itself. Similarly, a distinct and clear zone of inhibition was observed against *Aureobasidium pullalans NCIM 1049*, which confirms antifungal activity of Dhavana oil.

Results depict antimicrobial activity of neat Dhavana oil as well as retention of antibacterial property in encapsulated product. It can also be concluded that physical processing during encapsulation of Dhavana oil did not compromise the functional properties of the oil. The retained antibacterial activity of encapsulated Dhavana oil can be taken advantage of in various pharmaceutical, cosmetic, as well as food industries.

## 4. Conclusions

A stable emulsion of encapsulated Dhavana oil was obtained using modified starch gum, wherein the encapsulated product was obtained as free-flowing round or oval-shaped encapsulates with a moisture content of ~0.631%. Particle size of the encapsulated product was in the range of 5 to 236 µm, wherein 90% of them were <59.36 µm. The encapsulated product’s surface and total oil content were calculated to be 3.526% and 29.02%, respectively, with an encapsulation efficiency of 96.46%. Encapsulated Dhavana oil showed better stability than neat Dhavana oil with respect to % weight loss with increased temperature, which indicated that the encapsulated Dhavana oil is more stable in the modified starch gum matrix. Further, thermal degradation or % weight loss of neat and encapsulated Dhavana oil in TALC base was less than CaCO_3_, which indicated that TALC was a better base than CaCO_3_ and the product was more stable in TALC than CaCO_3_. Hydroxydavanone was found to be the active content that remained well protected under encapsulated form even after two months at 45 °C. Release studies confirmed that the loss of neat and encapsulated Dhavana oil in CaCO_3_ base was more than that in TALC base at 45 °C till two months. Compared to neat oil, the encapsulated Dhavana oil under open condition at 45 °C remained inside the gum matrix in both powder base TALC and CaCO_3_. FTIR analysis showed that Dhavana oil remained protected in an encapsulated format.

Dhavana oil showed antibacterial properties against *Escherichia coli*, *Staphylococcus aureus*, *Micrococcus* spp., *Bacillus subtilis*, *Serratia* spp., Klebsiella pneumoniae, *Streptococcus mutans*, *Salmonella typhi*, and the activity remained similar after conversion of oil to an encapsulated Dhavana oil. Dhavana oil showed antifungal properties against Aureobasidium pullulans NCIM 1049. MIC and MBC assay against *E. coli* and *S. aureus* confirm the broad-spectrum antibacterial activity of encapsulated product. The MIC of dhavana oil against *S. aureus* known around 6.4 mg/mL [21], whereas that of *E. coli* is unknown.

The encapsulated product can be added to a powder base like TALC or CaCO_3_, and the premix can be used for the final formulation. Therefore, our study is interesting from the perspective of various cosmetics formulations or detergent powders, cleaning powders, or similar formulations where we need to incorporate an essential oil or encapsulated essential oil. The encapsulation process helps to increase the shelf life as well as retain the antibacterial properties of Dhavana oil, posing as a better alternative to the use of neat Dhavana oil for long-term usage/applications.

## Figures and Tables

**Figure 1 molecules-27-07679-f001:**
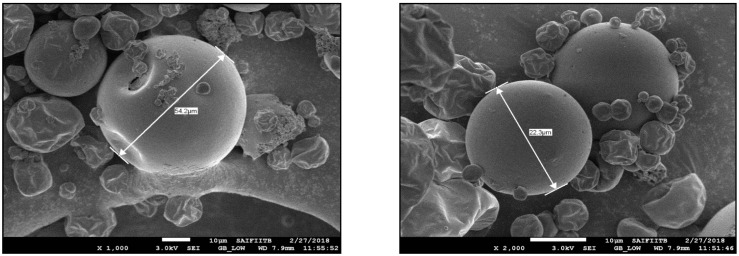
Scanning electron microscopic images on encapsulated Dhavana Oil.

**Figure 2 molecules-27-07679-f002:**
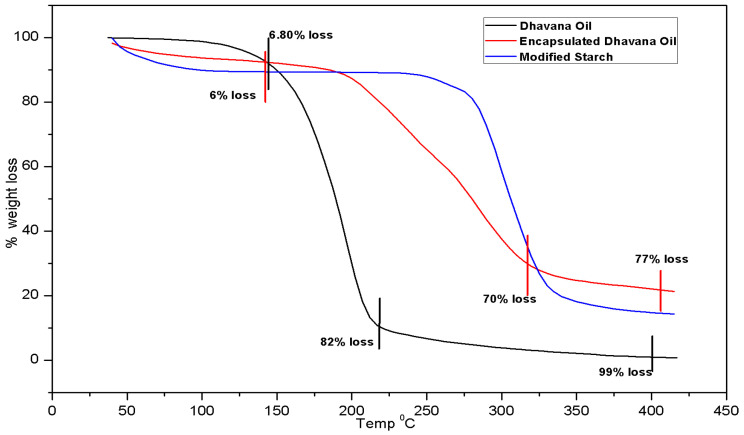
TGA analysis of the neat and encapsulated Dhavana oil.

**Figure 3 molecules-27-07679-f003:**
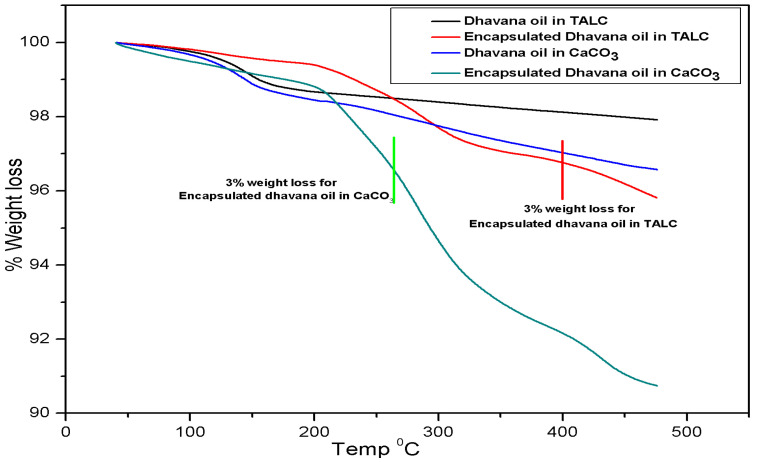
TGA analysis of the neat and encapsulated Dhavana oil in TALC and CaCO_3_.

**Figure 4 molecules-27-07679-f004:**
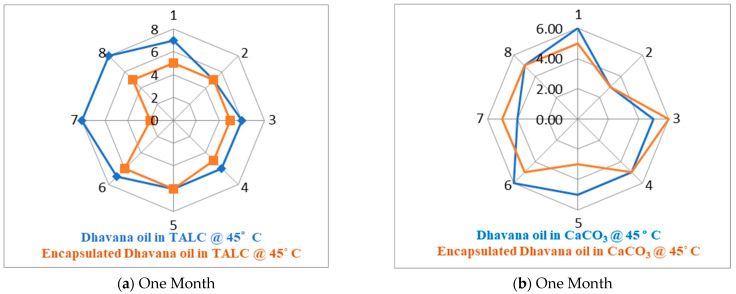
(**a**–**f**) Olfactory analysis of neat and encapsulated Dhavana oil in TALC and CaCO_3_ up to 3 months at 45 °C.

**Figure 5 molecules-27-07679-f005:**
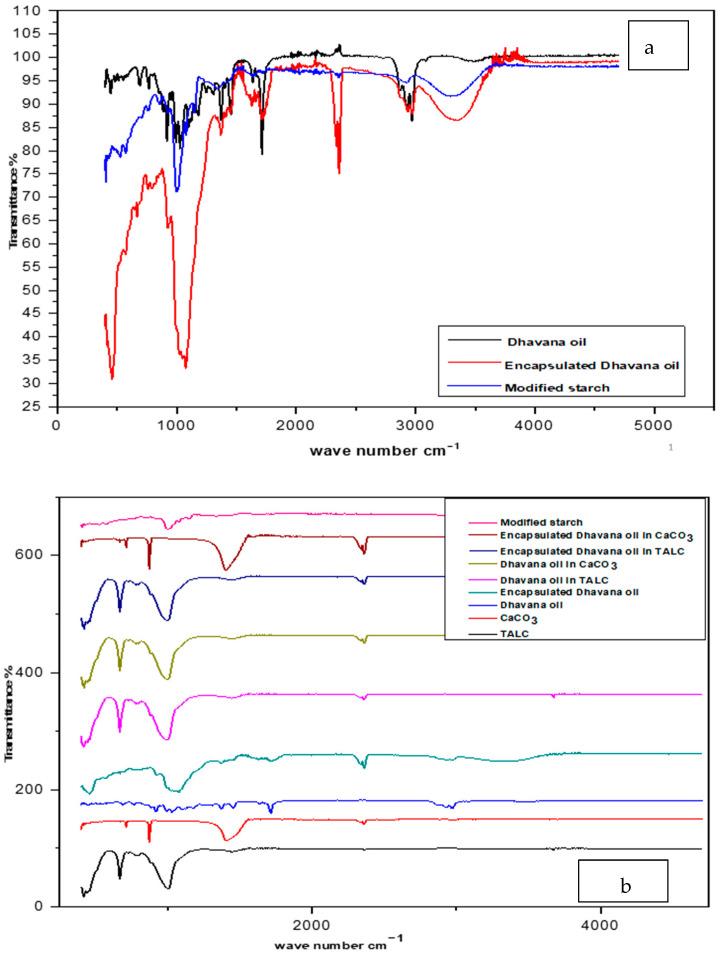
(**a**,**b**): FTIR analysis of neat and encapsulated Dhavana oil, Modified starch, TALC, CaCO_3_.

**Table 1 molecules-27-07679-t001:** Final formulation of preparation of Dhavana oil emulsion.

Sr.no.	Name of Ingredient	100% *w*/*w*
1	Citric Acid Anhydrous	2.00~5.00
2	Modified starch	50.00~55.00
3	Tri potassium citrate	3.00~8.00
4	Sucrose	5.00~12.00
5	Oil (Dhavana oil)	30.000
6	DM Water	150.000

**Table 2 molecules-27-07679-t002:** Parameters of the spray dryer.

Sr. No.	Parameter	Reading
1	Inlet temperature	160–170 °C
2	Outlet temperature	92 °C
3	Aspirator	45% (1350 rpm)
4	Vacuum	50
5	Air pressure	2 bar
6	Total time to complete a spray-drying	40 to 45 min.

**Table 3 molecules-27-07679-t003:** Physico-chemical parameters of neat Dhavana Oil.

Sr. No.	Test	Observation	Specification
1	Color	Brownish -Yellow viscous liquid	Brownish Yellow viscous liquid
2	Odor	Aromatic, Balsamic, Fruity, Woody, Sweet	Aromatic, Balsamic, Fruity, Woody, Sweet
3	Solubility	Clearly soluble in less than 1.5 volumes of 99% ethyl alcohol (Hyman grade)	Alcohol & Hexane soluble
4	Refractive index at 25 °C (Abbe refractometer Atago)	1.486	1.4794 to 1.4917
5	Specific gravity at 25 °C	0.9591	0.9394 to 0.9560
6	Acid value	2.95	≤3.5
7	Flashpoint Seta Multiflash (closed cup)	110 °C	≥97 °C
8	Boiling Point (capillary method)	190 °C	
9	GC-MS analysis (Firmenich)	Complete (Report as attached)	
10	Total ketones content as Davanone (GCMS-Firmenich)	56.00%	36 to 56.0%
11	Free Davanone as per GC	Passes	25 to 52.5%
12	Viscosity BF at 25 °C (100 rpm, spindle 3, BF RVT)	28 cps	N.A.
13	pH @1% solution of Dhavana oil in water.	4.88	Less than 7.00
14	Saponification Value	57.45 mg KOH/g	

**Table 4 molecules-27-07679-t004:** Particle size analysis of encapsulated Dhavana oil.

Sr. no	Test	Observation
1	10% particles below	6.71 µm
2	50% particles below	19.35 µm
3	90% of particles below	59.36 µm
4	100% particles below	236.78 µm

**Table 5 molecules-27-07679-t005:** Total oil, surface oil, and encapsulation efficiency of Modified Starch.

Sr. No.	Surface Oil (%)	Average Surface Oil (%)	Total Oil (%)	Average Total Oil (%)	Encapsulation Efficiency (%)	Average Encapsulation Efficiency (%)	Entrapment Efficiency (%)	Average Entrapment Efficiency
1	3.44	3.526%	28.92	29.02%	96.40%		88.10%	
2	3.57		29.17		97.23%	96.69%	87.76%	87.84%
3	3.57		28.94		96.46%		87.66%	

**Table 6 molecules-27-07679-t006:** Consolidated GCMS profile of neat and encapsulated Dhavana oil.

Sr.no	Component	Matrix	Duration	Retention Time	Area %
1	Neat Dhavana oil	TALC	One Month	44.0583	30.1827
2	Encapsulated Dhavana oil	TALC	One Month	44.0395	30.8049
3	Neat Dhavana oil	TALC	Two Months	44.0453	23.5616
4	Encapsulated Dhavana oil	TALC	Two Months	44.0364	26.7507
5	Neat Dhavana oil	CaCO_3_	One Month	43.1950	10.6000
6	Encapsulated Dhavana oil	CaCO_3_	One Month	43.1960	25.0000
7	Neat Dhavana oil	CaCO_3_	Two Months	43.1870	10.7100
8	Encapsulated Dhavana oil	CaCO_3_	Two Months	43.1870	24.5200

**Table 7 molecules-27-07679-t007:** Release of hydroxydhavanone from neat and encapsulated Dhavana oil.

Sr. No.	Component	Matrix	Duration	% Of Hydroxydhavanone
1	Encapsulated Dhavana Oil	TALC	Two months	26.75
2	Neat Dhavana oil	TALC	Two months	23.56
3	Encapsulated Dhavana Oil	TALC	One month	30.8
4	Encapsulated Dhavana Oil	CaCO_3_	Two months	24.52
5	Neat Dhavana oil	CaCO_3_	Two months	10.71
6	Encapsulated Dhavana Oil	CaCO_3_	One month	25

**Table 8 molecules-27-07679-t008:** Presence of active hydroxydhavanone from neat and encapsulated Dhavana oil in TALC and CaCO_3_ after 1 week in open condition (Petri dish) @ RT (22 °C) and 45 °C.

Sr.no.	Component	Matrix	Temperature	Occurrence of the Active Ingredient
001A	Neat Dhavana oil	TALC	RT	Present
001B	Neat Dhavana oil	TALC	45 °C	Absent
002A	Encapsulated Dhavana oil	TALC	RT	Present
002B	Encapsulated Dhavana oil	TALC	45 °C	Present
003A	Neat Dhavana oil	CaCO_3_	RT	Present
003B	Neat Dhavana oil	CaCO_3_	45 °C	Absent
004A	Encapsulated Dhavana oil	CaCO_3_	RT	Present
004B	Encapsulated Dhavana oil	CaCO_3_	45 °C	Present

**Table 9 molecules-27-07679-t009:** Comparison of intensity scores of neat Dhavana oil in TALC and CaCO_3_ after 1, 2 and 3 months@45 °C.

Respondent Number	Intensity Score	Intensity Score	Intensity Score	Intensity Score	Intensity Score	Intensity Score
	Dhavana Oil in TALC 1M	Dhavana Oil in CaCO3 1M	Dhavana Oil in TALC 2M	Dhavana Oil in CaCO3 2M	Dhavana Oil in TALC-3M	Dhavana Oil in CaCO3-3M
1	7	6	5	5	4	3
2	5	3	6	3	9	5
3	6	5	5	5	5	4
4	6	5	4	6	4	5
5	6	5	8	4	7	3
6	7	6	4	5	5	4
7	8	4	4	3	5	3
8	8	5	8	6	8	7
Total	53	39	44	37	47	34

**Table 10 molecules-27-07679-t010:** Results of neat and encapsulated Dhavana oil against bacterial and fungal strains in the form of a zone of inhibition obtained (mm) are tabulated below.

Name of Bacterial Strain	With Neat Dhavana Oil Zone Size (mm)			
	300 mg/mL	150 mg/mL	75 mg/mL	37.5 mg/mL
*S. aureus*	17 ± 0.2	14 ± 0.3	12 ± 0.1	-
*Micrococcus* spp.	18 ± 0.4	16 ± 0.2	13 ± 0.1	-
*E. coli*	15 ± 0.2	13 ± 0.1	11 ± 0.2	-
*Serratia* spp.	16 ± 0.2	13 ± 0.1	-	-
*K. pneumoniae*	16 ± 0.3	13 ± 0.4	-	-
*B. subtilis*	12 ± 0.1	-	-	-
*S. typhi*	15 ± 0.3	-	-	-
*S. mutans*	14 ± 0.2	13 ± 0.1	11 ± 0.2	-
*P. aeruginosa*	-	-	-	-
Name of Bacterial Strain	With Encapsulated Dhavana oil Zone size (mm)			
	300 mg/mL	150 mg/mL	75 mg/mL	37.5 mg/mL
*E. coli*	16 ± 0.2	13 ± 0.2	-	-
*S. aureus*	16 ± 0.4	13 ± 0.3	-	-
*Micrococcus* spp.	15 ± 0.1	13 ± 0.2	-	-
*S. mutans*	18 ± 0.3	16 ± 0.1	15 ± 0.2	10 ± 0.1
Name of Fungal Strain	With neat Dhavana oil Zone size (mm)			
	200 μL			
*Aureobasidium pullulans*	18.5			
*NCIM 1049*				

## Data Availability

Data available on request from the authors.

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
