# Peer review of "Physico-Chemical and Antimicrobial Efficacy of Encapsulated Dhavana Oil: Evaluation of Release and Stability Profile from Base Matrices"

_molecules, 2022, doi:10.3390/molecules27227679_

Round 1
Reviewer 1 Report
Promising results were obtained after a study of the encapsulation of Dhavana essential oil in modified starch using CaCO3 and Talc as base matrices. The manuscript presents a compilation of important information on volatile release and preservation of antimicrobial activities after encapsulation. It is duly formulated and indicated for publication. Authors need to check the manuscript's suitability for the journal's standards.
The topic addressed by the authors has technological relevance since the essential oil of Artemisia pallen is produced on an industrial scale. Data on the encapsulation of volatiles from this plant is not found in the literature. The authors proved the greater thermal stability, gradual release of volatiles, and preservation of the oil's antibacterial activity after encapsulation. However, the work was incomplete, as the antifungal evaluation of the encapsulated and antibacterial and antifungal assessment of encapsulated oil after the addition of CaCO3 and TALC were not reported.
Regarding the methodology:
Item 2.2. Equipment may be removed and used equipment may be mentioned in the test description. I suggest replacing this item with 2.2 Analytical Methodology, which must include all the methods mentioned in Table 1
Item 2.3. Experimental Details must only describe the methodology of the trials without presenting the results (Table 1). This table should be in the results.
The results lacked information about the antifungal properties of the encapsulated oil. I need to add this information.
The results of antibacterial and antifungal activity after application of powder bases: CaCO3 and TALC were not presented.
References are adequate, but authors must check that they are all within the journal's standards. Parentheses must not be placed after the number.
Author Response
We thank our reviewers for reviewing our manuscript and giving us fruitful suggestions. The manuscript has been revised as per reviewer's comments.

Reviewer 2 Report
The authors have done a great job. Still, I have the following considerations:
The abstract is too long. It is recommended to shorten it.
The authors make a good explanation of the results, but they need to be discussed further by comparing their data with those in the bibliography.
The technique used for encapsulation is dispersion by drying, 92ºC is used as outlet temperature, obtaining an encapsulation efficiency of 96%. How is this possible, if the essential oils evaporate above 70ºC.
The authors indicate that there are no differences in the antimicrobial activity of the pure and encapsulated oil. To what is due?. The encapsulation should improve the MIC and the MBC, needing a lower concentration of the essential oil to reach them.
Which of the essential oil components is responsible for the antimicrobial activity?
Once the essential oil is complexed, regardless of the method used, it has been proven that components of the oil are actually encapsulated.
Because the authors take Hydroxydavanone as a reference to see the release capacity of the oil, if this molecule appears at a very low concentration in the oil.
Author Response

(The authors gave the same response as above.)

Reviewer 3 Report
This article is devoted to the study of the physicochemical and antimicrobial efficacy of encapsulated dhavana oil. The article is understandable. The relevance is beyond doubt. There are some points to fix:
1. There is not always a logical connection between paragraphs. Pay attention to this.
2. "3.1." comparison with other literature data is necessary.
3. "Table 5." can be described in more detail.
4. You can cite the work: 10.3390/molecules27186129.
5. "3.3.4. FTIR analysis" described too succinctly. Pay attention to subtle effects. In addition, references to the literature will also be helpful in this context.
6. Antimicrobial activity is also strongly compared with data from the literature. This is with standard substances and with other essential oils that have this activity.
Author Response

(The authors gave the same response as above.)

Round 2
Reviewer 3 Report
Acceped